# Noise-Robust Semi-Supervised Learning for Distantly Supervised Relation Extraction

**Xin Sun**[1,2], **Qiang Liu**[2,3,*] **Shu Wu**[2,3], **Zilei Wang**[1], **Liang Wang**[2,3]

[1]University of Science and Technology of China
[2]CRIPAC & MAIS, Institute of Automation, Chinese Academy of Sciences
[3]University of Chinese Academy of Sciences

sunxin000@mail.ustc.edu.cn, qiang.liu@nlpr.ia.ac.cn

## Abstract

Distantly supervised relation extraction (DSRE) aims to extract relational facts from texts but suffers from noisy instances. To mitigate the influence of noisy labels, current methods typically use the Multi-Instance-Learning framework to extract relations for each bag. However, these approaches are not capable of extracting relation labels for individual sentences. Several studies have focused on sentence-level DSRE to solve the above problem. These studies primarily aim to develop methods for identifying noisy samples and filtering them out to mitigate the impact of noise. However, discarding noisy samples directly leads to the loss of useful information. To this end, we propose SSLRE, a novel **S**emi-**S**upervised-**L**earning **R**elation **E**xtraction framework for sentence-level DSRE. We discard only the labels of the noisy samples and utilize these instances without labels as unlabeled samples. Our SSLRE framework utilizes a weighted K-NN graph to select confident samples as labeled data and the rest as unlabeled. We then design a robust semi-supervised learning framework that can efficiently handle remaining label noise present in the labeled dataset, while also making effective use of unlabeled samples. Based on our experiments on two real-world datasets, the SSLRE framework we proposed has achieved significant enhancements in sentence-level relation extraction performance compared to the existing state-of-the-art methods. Moreover, it has also attained a state-of-the-art level of performance in bag-level relation extraction with ONE aggregation strategy.

## 1 Introduction

Relation extraction (RE) is a fundamental process for constructing knowledge graphs, as it aims to predict the relationship between entities based on their context. However, most supervised RE

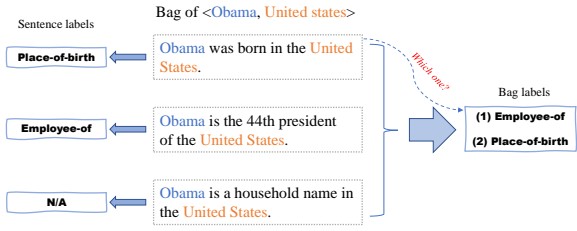

Figure 1: Bag-level RE maps a bag of sentences to bag labels. Sentence-level RE maps each sentence to a specific relation.

techniques require extensive labeled training data, which can be difficult to obtain manually. To address this issue, Distant Supervision (DS) was proposed (Mintz et al., 2009) to automatically generate labeled text corpus by aligning plain texts with knowledge bases (KB). For instance, if a sentence contains both the subject ($s$) and object ($o$) of a triple $\langle s, r, o \rangle$ ($\langle subject, relation, object \rangle$), then the DS method considers $\langle s, r, o \rangle$ as a valid sample for that sentence. Conversely, if no relational triples apply, then the sentence is labeled as "NA".

Distantly supervised datasets usually face high label noise in training data due to the annotation process. To mitigate the impact of noisy labels caused by distant supervision, contemporary techniques (Lin et al., 2016; Alt et al., 2019; Chen et al., 2021b; Li et al., 2022; Dai et al., 2022) usually employ multi-instance-learning (MIL) framework or modify MIL to train the relation extraction model.

Although MIL-based techniques can identify bag relation labels, they are not proficient in precisely mapping each sentence in a bag with explicit sentence labels (Feng et al., 2018; Jia et al., 2019; Ma et al., 2021). Several studies have focused on improving sentence-level DSRE and have empirically demonstrated the inadequacy of bag-level methods on sentence-level evaluation. (Feng et al., 2018; Qin et al., 2018) apply reinforcement learning to train a sample selector. (Jia et al., 2019) iden-

---
*Corresponding author.

tify confident samples by frequent patterns. (Ma et al., 2021) utilizes negative training to separate noisy data from the training data.

However, these methods has two main issues: (1) These works simply discard all noisy samples and train a relation extraction model with selected samples. However, filtering out noisy samples directly results in the loss of useful information. (Gao et al., 2021a) notes that the DSRE dataset has a noise-rate exceeding 50%. Discarding all these samples would result in a significant loss of information. (2) The confident samples selection procedure is not impeccable, and there may still exist a small amount of noise in the chosen confident samples. Directly training a classifier in the presence of label noise is known to result in noise memorization.

To address the two issues, this work proposes a novel semi-supervised-learning framework for sentence-level DSRE, First, we construct a K-NN graph for all samples using the hidden features. Then, we identify confident samples from the graph structure and consider the remaining samples as noisy. For issue (1): Our method discards only the noisy labels and treats corresponding samples as unlabeled data. We then utilize this unlabeled data by pseudo labeling within our robust semi-supervised learning framework to learn a better feature representation for relation. For issue (2): Despite our initial selection of confident samples, there may still be noise in the labeled dataset. we have developed a noise-robust semi-supervised learning framework that leverages mixup supervised contrastive learning to learn from the labeled dataset and curriculum pseudo labeling to learn from the unlabeled dataset.

To summarize the contribution of this work:

- We propose a noise-robust Semi-Supervised-Learning framework SSLRE for DSRE task, which effectively mitigate the impact of noisy labels.

- We propose to use graph structure information (weighted K-NN) to identify the confident samples and effectively convert noisy samples as useful training data by utilizing pseudo labeling.

- The proposed framework achieves significant improvement over previous methods in terms of both sent-level and bag-level relation extraction performance.

## 2 Related work

### 2.1 Distantly Supervised Relation Extraction

Relation extraction (RE) is a fundamental pro- cess for constructing knowledge graphs(Zhang et al., 2023a; Xia et al., 2022; Zhang et al., 2023b). To generate large-scale auto-labeled data without human effort, (Mintz et al., 2009) first use Distant Supervision to label sentences mentioning two entities with their relations in KGs, which inevitably brings wrongly labeled instances. To tackle the predicament of noise, most of the existing studies on DSRE are founded on the multi-instance learning framework. This approach is leveraged to handle noisy sentences in each bag and train models by capitalizing on the constructed trustworthy bag-level representations. Usually, these methods employ attention mechanisms to assign less weights to the probable noisy sentences in the bag(Lin et al., 2016; Han et al., 2018b; Alt et al., 2019; Shen et al., 2019; Ye and Ling, 2019; Chen et al., 2021a; Li et al., 2022), apply adversarial training or reinforcement learning to remove the noisy sentences from the bag (Zeng et al., 2015; Qin et al., 2018; Shang and Wei, 2019; Chen et al., 2020; Hao et al., 2021). However, the studies (Feng et al., 2018; Jia et al., 2019; Ma et al., 2021; Gao et al., 2021a) indicate that the bag-level DSRE methods are ineffective for sentence-level relation extraction.

This work focuses on extracting relations at the sentence level, (Feng et al., 2018) applied reinforcement learning to identify confident instances based on the reward of noisy labels. (Jia et al., 2019) involve building initial reliable sentences based on several manually defined frequent relation patterns. (Ma et al., 2021) assigning complementary labels that cooperate with negative training to filter out noisy instances. Unlike previous studies, our method only discard noisy labels and keep the unlabeled samples. We use pseudo labeling to effectively utilize unlabeled samples, which helps to learn better representation.

### 2.2 Semi-Supervised-Learning

In SSL, a portion of training dataset is labeled and the remaining portion is unlabeled. SSL has seen great progress in recent years. Since (Bachman et al., 2014) proposed a consistency regularization based method, many approaches have migrated it into the semi-supervised learning field. MixMatch (Berthelot et al., 2019) proposes to combine consistency regularization with entropy minimization.

Mean Teacher (Tarvainen and Valpola, 2017) and Dual Student (Ke et al., 2019) are also based on consistent learning, aiming for the same outputs for different networks. Recently, FixMatch (Sohn et al., 2020) provides a simple yet effective weak-to-strong consistency regularization framework. Flexmatch (Zhang et al., 2021) provides curriculum pseudo pesudo labels learning to combat the imbalance of pseudo labels.

Our SSLRE framework differs from these frameworks in two main ways. Firstly, our labeled dataset still contains a small amount of noise due to the fact that confident sample identification cannot achieve perfection. Therefore, we utilize mixup contrastive supervised learning to combat this noise. Secondly, current SSL methods generate and utilize pseudo labels with the same head, which causes error accumulation during the training stage. To address this issue, we propose utilizing a pseudo classifier head, which decouples the generation and utilization of pseudo labels by two parameter-independent heads to avoid error accumulation.

## 2.3 Learning with Noisy Data

In both computer vision and natural language processing, learning with noisy data is a widely discussed problem. Existing approaches include but not limit to estimating the noise transition matrix (Chen and Gupta, 2015; Goldberger and Ben-Reuven, 2016), leveraging a robust loss function (Lyu and Tsang, 2019; Ghosh et al., 2017; Liu and Guo, 2019), introducing regularization (Liu et al., 2020; Iscen et al., 2022), selecting noisy samples by multi-network learning or multi-round learning (Han et al., 2018a; Wu et al., 2020), re-weighting examples (Liu and Tao, 2014), generating pseudo labels (Li et al., 2020; Han et al., 2019), and so on. In addition, some advanced state-of-the-art methods combine serveral techniques, e.g., Dividemix (Li et al., 2020) and ELR+ (Liu et al., 2020).

In this paper, we address the issue of noisy labels in distant relation extraction. Our approach first involves constructing a K-NN graph to identify confident samples based on their graph structure, and then use noise-robust mixup supervised contrastive learning to train with the labeled samples.

## 3 Methodology

To achieve sentence-level relation extraction in DSRE, we propose a framework called SSLRE, which consists of two main steps. Firstly, we select confident samples from the distantly supervised dataset using a weighted K-NN approach built by all sample representations. We use the selected samples as labeled data and the remaining samples as unlabeled data (as detailed in Section 3.1). Secondly, we employ our robust Semi-Supervised Learning framework to learn from the Semi-Supervised datasets (as described in Section 3.2). Appendix A delineates the full algorithm.

Specifically, we denote the original dataset in this task as $\tilde{\mathcal{D}} = \{(s_1, \tilde{y}_1), (s_2, \tilde{y}_2), \cdots, (s_N, \tilde{y}_N)\}$, where $\tilde{y}_i \in \{1, \cdots, C\}$ is the label of the $i$th input sentence $s_i$. The labeled dataset (identified confident samples) is denoted as $\mathcal{X} = \{(x_1, y_1), (x_2, y_2), \cdots, (x_n, y_n)\}$ and the unlabeled dataset (the noisy samples without labels) is denoted as $\mathcal{U} = \{u_1, u_2, \cdots, u_m\}$, where $m + n = N$.

### 3.1 Confident Samples Identification with Weighted K-NN

Our Semi-Supervised-Learning module requires us to divide the noisy dataset into a labeled dataset and an unlabeled dataset. Inspired by (Lee et al., 2019; Bahri et al., 2020), we utilize neighborhood information of the hidden feature spaces to identify confident samples We employ supervised contrastive learning to warm up our model and obtain the representations for all instances. It is noteworthy that deep neural networks tend to initially fit the training data with clean labels during an early training phase, prior to ultimately memorizing the examples with false labels (Arpit et al., 2017; Liu et al., 2020). Consequently, we only warm up our model for a single epoch. Given two sentences $s_i$ and $s_j$, we can obtain their low-dimensional representations as $z_i = \theta(s_i)$ and $z_j = \theta(s_j)$, where $\theta$ is the sentence encoder. We then calculate their representation similarity using the cosine distance

$$d(z_i, z_j) = \frac{z_i z_j^T}{\|z_i\| \|z_j\|}. \quad (1)$$

Then, we build a weighted K-NN graph for all samples based on the consine distance. To quantify the agreement between $s_i$ and $\tilde{y}_i$, We first use the label distribution in the $K$-neighborhood to approximate clean posterior probabilities,

$$\hat{q}_c(s_i) = \frac{1}{\sum_{s_k \in \mathcal{N}_i} d(z_i, z_k)} \sum_{s_k \in \mathcal{N}_i} d(z_i, z_k) \cdot \mathbb{1}(\tilde{y}_k = c) \quad (2)$$

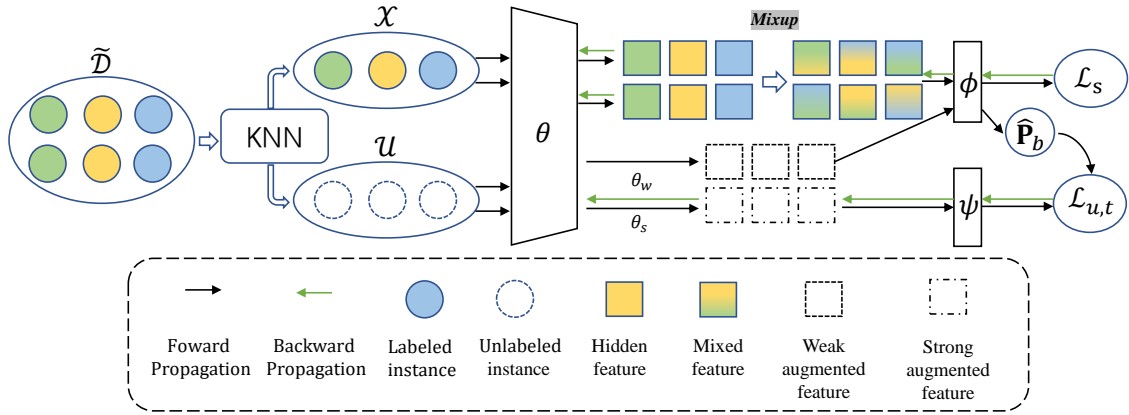

Figure 2: An overview of the proposed framework, SSLRE. $\hat{\mathcal{D}}, \mathcal{X}, \mathcal{U}$ denote the original noisy dataset, labeled dataset and unlabeled dataset. $\theta$ indicates the encoder. $\theta_w$ and $\theta_s$ mean forward with lower and higher dropout rate, respectively. $\phi$ and $\psi$ are classifier head and pseudo classifier head. $\mathcal{L}_s$ is the mixup supervised contrastive loss defined in Eq. (12), and $\mathcal{L}_{u,t}$ is the unsupervised loss defined in Eq. (6).

where $\mathcal{N}_i$ represents the set of $K$ closest neighbors to $s_i$. We then use the cross-entropy loss $\ell$ to calculate the disagreement between $\hat{\mathbf{q}}_c(s_i)$ and $\tilde{y}_i$. Denoting the set of confident examples belonging to the $c$-th class as $\mathcal{X}_c$, we have

$$\mathcal{X}_c = \{s_i, \tilde{y}_i \mid \ell(\hat{\mathbf{q}}(s_i), \tilde{y}_i) < \gamma_c, \tilde{y}_i = c\}, c \in [C], \tag{3}$$

where $\gamma_c$ is a threshold for the $c$-th class, which is dynamically defined to ensure a class-balanced set of identified confident examples. To achieve this goal, we use the $\alpha$ fractile of per-class agreements between the original label $\tilde{y}_i$ and $\max_c \hat{\mathbf{q}}_c(s_i)$ across all classes to determine how many examples should be selected for each class, i.e. $\sum_{i=1}^n = \mathbb{1}(\max_{c'} \hat{\mathbf{q}}_{c'}(s_i) = \tilde{y}_i) \cdot (\tilde{y}_i = c), c \in [C]$. Finally, we can get the labeled set and unlabeled set as

$$\begin{aligned} \mathcal{X} &= \cup_{c=1}^C \mathcal{X}_c \\ \mathcal{U} &= \{s_i \mid (s_i, \tilde{y}_i) \in \tilde{\mathcal{D}} \setminus \mathcal{X}\}. \end{aligned} \tag{4}$$

## 3.2 Noise-Robust Semi-Supervised learning

Despite selecting confident samples from the distantly supervised dataset, there still remains a small amount of noise in the labeled dataset. Naively training a classifier in the presence of label noise leads to noise memorization (Liu et al., 2020), which degrades the performance. We propose a noise-robust semi-supervised learning framework to mitigate the influence of remaining noise.

### 3.2.1 Data Augmentation with Dropout

Inspired by SimCSE (Gao et al., 2021b), we augment training samples by embedding processing. In particular, We obtain different embeddings of a sentence by applying dropout during the forward process. Additionally, we propose using a high dropout rate for strong augmentation and a low dropout rate for weak augmentation. The sentence encoder is denoted as $\theta$, with forward propagation using a high dropout rate denoted as $\theta_s$ and forward propagation using a low dropout rate denoted as $\theta_w$.

### 3.2.2 Unsupervised Learning with Pseudo Labeling

In this part, we propose two modules to learn from the unlabeled dataset: (1) To generate and utilize pseudo labels independently, we propose pseudo classifier head. (2) Utilize Curriculum Pseudo Labeling to perform consistent learning while combating the unbalance of the generated pseudo labels.

**Pseudo classifier head**: pseudo labeling is one of the prevalent techniques in semi-supervised learning. The existing approaches generate and utilize pseudo labels with the same head. However, this may cause training bias, ultimately amplifying the model's errors as self-training continues. (Wang et al., 2022). To reduce this bias when using pseudo labels, we propose utilizing a two-classifier model consisting of an encoder $\theta$ with both a classifier head $\phi$ and a pseudo classifier head $\psi$. We optimize the classifier head $\phi$ using only labeled samples and without any unreliable pseudo labels from unlabeled samples. Unlabeled samples are used solely for updating encoder $\theta$ and pseudo classifier head $\psi$. In particular, the classifier head $\phi$ generates pseudo labels $(\phi \circ \theta_w)(u_b)$ for unlabeled samples (which have

no gradient), the loss of unlabeled samples is calculated by $\ell((\psi \circ \theta_s)(u_b), (\phi \circ \theta_w)(u_b))$, where $\ell$ denotes cross entropy loss. This decouples the generation and utilization of pseudo labels by two parameter-independent heads to mitigate error accumulation.

**Curriculum Pseudo Labeling**: Due to the highly unbalanced dataset, using a constant cut-off $\tau$ for all classes in Pseudo labeling results in almost all selected samples (those with confidence greater than the cut-off) being labeled as 'NA', which is the dominant class.

Inspired by Flexmatch (Zhang et al., 2021), we use Curriculum Pseudo Labeling (CPL) to combat unbalanced pseudo labels. We first generate pseudo labels for iteration $t$

$$\hat{\mathbf{p}}_b = (\phi_t \circ \theta_{w,t})(u_b), \qquad (5)$$

These labels are then used as the target of strongly-augmented data. The unsupervised loss term has the form as

$$\mathcal{L}_{u,t} = \frac{1}{\mu B} \sum_{b=1}^{\mu B} \mathbb{1} \left( \max(\hat{\mathbf{p}}_b) > \mathcal{T}_t(\arg\max(\hat{\mathbf{p}}_b)) \right) \\ \cdot \ell((\psi_t \circ \theta_{s,t})(u_b), \hat{\mathbf{p}}_b), \qquad (6)$$

where

$$\mathcal{T}_t(c) = \frac{\sigma_t(c)}{\max_c \sigma_t} \tau, \qquad (7)$$

and $\sigma_t(c)$ represents the numbers of the samples whose predictions fall into class $c$ and above the threshold, formulated as

$$\sigma_t(c) = \sum_{n=1}^{N} \mathbb{1}(\max \hat{\mathbf{p}}_b > \tau) \cdot \mathbb{1}(\arg\max \hat{\mathbf{p}}_b = c). \qquad (8)$$

### 3.2.3 Mixup Supervised Contrastive Learning

We target learning robust relation representation in the presence of remaining label noise. In particular, we adopt the contrastive learning approach and randomly sample $N$ sentences and inference the sentences twice with same dropout rate to get two view. Then we normalize the embedding by $L_2$ normalization. The resulting minibatch $\{z_i, y_i\}_{i=1}^{2N}$ consists of $2N$ normalized sentence embedding and corresponding labels. We perform supervised contrastive learning on labeled samples

$$\mathcal{L}_i(z_i, y_i) = \frac{1}{2N_{y_i} - 1} \sum_{j=1}^{2N} \mathbb{1}_{i \neq j} \mathbb{1}_{y_i = y_j} \\ \cdot \log \frac{\exp(z_i \cdot z_j / \tau)}{\sum_{r=1}^{2N} \mathbb{1}_{r \neq i} \exp(z_i \cdot z_r / \tau)}. \qquad (9)$$

To make representation learning robust, we add Mixup (Berthelot et al., 2019) to supervised contrastive learning. Mixup strategies have demonstated excellent performance in classification frameworks and have futher shown promising results to prevent label noise memorization. Inspired by this success, we propose mixup supervised contrastive learning, a novel adaptation of mixup data augmentation for supervised contrastive learning. Mixup performs convex combinations of pairs of samples as

$$x_i = \lambda x_a + (1 - \lambda) x_b, \qquad (10)$$

where $\lambda \in [0,1] \sim Beta(\alpha_m, \alpha_m)$ and $x_i$ denotes the training example that combines two mini-batch examples $x_a$ and $x_b$. A linear relation in the contrastive loss is imposed as

$$\mathcal{L}_i^{MIX} = \lambda \mathcal{L}_a(z_i) + (1 - \lambda) \mathcal{L}_b(z_i), \qquad (11)$$

where $\mathcal{L}_a$ and $\mathcal{L}_b$ have the same form as $\mathcal{L}_i$ in Eq. (9).

The supervised loss is the sum of Eq. (11) for each mixed instance:

$$\mathcal{L}_s = \sum_i^{2N} \mathcal{L}_i^{MIX}. \qquad (12)$$

Mixup supervised contrastive learning helps to learn a robust representation for relations, but it cannot map the representation to a class. To learn the map function from the learned representation to relation class, classification learning using cross entropy loss is also employed as

$$\mathcal{L}^{CLS} = \sum_{(x_i, y_i) \in \mathcal{X}} \mathcal{L}_i^{cls}(x_i) = \sum_{(x_i, y_i) \in \mathcal{X}} \ell((\phi \circ \theta)(x_i), y_i). \qquad (13)$$

### 3.3 Training Objective

Combining the above analyses, the total objective loss is

$$\mathcal{L} = \mathcal{L}_s + \lambda_u \mathcal{L}_{u,t} + \lambda_c \mathcal{L}^{CLS}. \qquad (14)$$

## 4 Experiments

### 4.1 Datasets

We evaluate our SSLRE framework on three DSRE datasets, including NYT10 (Riedel et al., 2010), NYT10m (Gao et al., 2021a), and wiki20m (Gao et al., 2021a).

**NYT10** dataset is created by aligning information from FreeBase with the New York Times (NYT)

corpus. However, (1) it contains many duplicated instances in the dataset; (2) there is no public validataion set; (3) The distantly supervised test set is quite noisy since the anotated errors. (Gao et al., 2021a) notes that 53% samples of the NYT10 test set are wrongly labeled. We only use it to perform held-out evaluation experiment in Table 5 with some strong baselines.

**NYT10m** build a manually annotated test set for NYT10 for more accurate evaluation. Futhermore, it removes all the duplicated instances and create a new relation ontology by merging semantically similar relations and delete the relations that only show up in train set or the test set.

**wiki20m** is a processed version of Wiki20 (Han et al., 2020), which is constructed by aligning the English Wikipedia corpus with Wikidata (Vrandečić and Krötzsch, 2014). It has a manually annotated test set for evaluation.

### 4.2 Evaluation and Parameter Settings

To guarantee the fairness of evaluation. We take both sentence-level evaluation and bag-level evaluation in our experiments. Further details of the evaluation methods are available in the appendix C. To achieve bag-level evaluation under sentence-level training, we use at-least-one (**ONE**) aggregation stragy (Zeng et al., 2015), which first predicts relation scores for each sentence in the bag, and then takes the highest score for each relation. The details of the hyper-parameters are available in the appendix D.

### 4.3 Baselines

In order to prove the effectiveness of the SSLRE, we compare our method with state-of-the-art methods sentence-level DSRE framework and bag-level DSRE framwork.

For bag-level methods baselines, RESIDE (Vashishth et al., 2018) exploits the information of entity type and relation alias to add a soft limitation for DSRE. DISTRE (Alt et al., 2019) combines the selective attention to its Transformer-based model. CIL (Chen et al., 2021a) utilize contrastive instance learning under MIL framwork, HiCLRE (Li et al., 2022) propose hierarchical contrastive learning framwork, PARE (Rathore et al., 2022) propose concatenate all sentences in the bag to attend every token in the bag. Besides, we combine Bert with different aggregation strategies: **ONE**, which is mentioned in section 4.2; **AVG** averages the representations of all the sentences in the bag;

**ATT** (Lin et al., 2016) produces bag-level representation as a weighted average over embeddings of sentences and determines weights by attention scores between sentences and relations.

For sentence-level baselines, RL-DSRE (Feng et al., 2018) apply reinforcement learning to train sample selector by feedback from the manually designed reward function. ARNOR (Jia et al., 2019) selects the reliable instances based on reward of attention score on the selected patterns. SENT (Ma et al., 2021) filters noisy instances based on negative training.

### 4.4 Results

We first evaluate our SSLRE framework in the NYT10m and WIKI20m dataset. Table 1 shows the overall performance in terms of sentence-level evaluation. From the results, we can observe that (1) Our SSLRE framework demonstrates superior performance on both datasets, surpassing all strong baseline models significantly in terms of F1 score. In comparison to the most robust baseline models in two distantly supervised datasets, SSLRE displays a significant enhancement in performance (i.e., +6.3% F1 and +3.4% F1). (2) The current sentence-level DSRE models (SENT, ARNOR) fail to outperform the state-of-the-art MIL-based techniques in terms of F1 score on the aforementioned datasets. This could be attributed to the loss of information resulting from the elimination of samples. Unlike the MIL-based methods that employ all samples for training, these models only utilize selected samples. Moreover, the selection procedure may not always be reliable. (3) The performance of state-of-the-art MIL-based methods is not substantially superior to that of the Bert baseline. This suggests that the MIL modules, which are specifically crafted for this task, do not exhibit significant effectiveness when evaluated at the sentence level.

Table 2 presents the results of bag-level evaluation of SSLRE with **ONE** strategy on NYT10m and WIKI20m datasets. We compared our SSLRE framework with state-of-the-art methods for bag-level relation extraction, and found that our approach outperformed all strong baselines. Specifically, SSLRE achieved a 5.5% improvement in AUC compared to the best baseline model PARE, and a 10.1% improvement in micro F1 score compared with best baseline model HICLRE on the NYT10m dataset. Despite being trained without a

| Models | NYT10m | | | wiki20m | | |
|---|---|---|---|---|---|---|
| | $\mu$Prec. | $\mu$Rec. | $\mu$F1 | $\mu$Prec. | $\mu$Rec. | $\mu$F1 |
| Bert-ATT | 49.1 | 58.2 | 53.3 | 71.3 | 77.8 | 74.4 |
| Bert-ONE | 51.0 | 60.7 | 55.5 | 72.8 | 73.2 | 73.0 |
| Bert-AVG | 52.0 | 55.8 | 53.9 | 80.7 | 76.9 | 78.7 |
| RESIDE | 45.5 | 48.7 | 47.0 | 69.9 | 72.3 | 70.9 |
| DISTRE | 55.1 | 51.0 | 52.9 | 80.3 | 73.6 | 76.8 |
| CIL | **58.1** | 49.3 | 53.4 | 81.8 | 73.1 | 77.0 |
| HiCLRE | 54.6 | 60.7 | 57.5 | 81.6 | 74.6 | 77.9 |
| PARE | 53.1 | 58.1 | 55.4 | 77.3 | 74.2 | 75.7 |
| RL-DSRE | 48.6 | 57.1 | 52.5 | 69.9 | 74.3 | 72.0 |
| ARNOR | 53.2 | 59.1 | 55.9 | 76.8 | 76.9 | 76.8 |
| SENT | 57.2 | 56.3 | 56.7 | 79.8 | 78.2 | 78.9 |
| SSLRE | 57.4 | **72.0** | **63.8** | **81.9** | **81.0** | **81.5** |

Table 1: Sentence-level evaluation results on NYT10m and wiki20m. Bold and underline indicate the best and the second best scores.

| Models | NYT10m | | | wiki20m | | |
|---|---|---|---|---|---|---|
| | AUC | $\mu$F1 | Macro_F1 | AUC | $\mu$F1 | Macro_F1 |
| Bert-ATT | 49.5 | 52.9 | 24.5 | 88.0 | 80.9 | 80.7 |
| Bert-ONE | 56.7 | 54.1 | 35.7 | 89.9 | 81.3 | 82.0 |
| Bert-AVG | 57.1 | 56.2 | 33.9 | 88.9 | 82.6 | 81.1 |
| RESIDE | 36.8 | 44.2 | 11.2 | 80.5 | 75.1 | 74.2 |
| HiCLRE | 61.0 | 61.2 | 32.0 | 89.1 | 82.3 | 81.1 |
| CIL | 57.4 | 59.6 | 29.4 | 89.3 | 81.8 | 82.4 |
| PARE | 61.1 | 59.8 | **37.2** | 90.3 | 83.2 | 82.6 |
| SSLRE-**ONE** | **66.5** | **71.3** | 36.9 | **91.6** | **83.3** | **84.1** |

Table 2: Bag-level evaluation results on nyt10m and wiki20m. SSLRE-**ONE** represents the SSLRE with **ONE** aggregation strategy

MIL framework, our SSLRE framework achieves state-of-the-art performance on bag-level relation extraction. This finding suggests that sentence-level training can also yield excellent results on bag-label prediction. This observation is also consistent with (Gao et al., 2021a; Amin et al., 2020).On the wiki20m dataset, we note a consistent improvement on as well, although it is not as evident as in the case of NYT10m. We surmise that this could be attributed to the fact that the wiki20m dataset is relatively less noisy when compared to NYT10m.

We also compared our framework to several strong baselines using held-out evaluation on the NYT10 dataset, which is detailed in appendix B.

### 4.5 Ablation Study

We conducted ablation study experiments on the NYT10m dataset to assess the effectiveness of different modules in our SSLRE framework. We specifically removed each of the argued contributions one by one to evaluate their effectiveness. For unsupervised learning part, we remove the pseudo classifier head and CPL one at a time. For supervised learning part, we switch from mixup supervised learning to supervised contrastive learning and cross entropy as our new objective. In terms of confident samples identifications methods, we alternate between using random (randomly selection) and NLI-based selection instead of our K-NN method. The NLI method involves performing zero-shot relation extraction through Natural Lan-

guage Inference (NLI)(Sainz et al., 2021), then identify the confident samples based on the level of agreement between the distant label and NLI soft label.

| Methods | | Prec. | Rec. | F1 |
|---|---|---|---|---|
| SSLRE | | 57.4 | 72.0 | 63.8 |
| Unsupervised Learning | w/o pseudo head | 56.2 | 70.6 | 62.5 |
| | w/o CPL | 50.8 | 69.6 | 58.7 |
| Supervised Learning | SupCon | 55.4 | 67.5 | 60.9 |
| | CE | 51.8 | 70.3 | 59.6 |
| Conf-Samples Identification | random | 58.6 | 56.0 | 57.3 |
| | NLI | 56.4 | 65.3 | 60.5 |

Table 3: Ablation study of SSLRE on NYT10m

Table. 3 shows the ablation study results. We conclude that (1) Unsupervised learning apart effectively utilize the unlabeled samples. Removing pseudo classifier head and CPL leads to a decrease of 1.3% and 5.1% on micro-F1, respectively. (2) When dealing with noisy labeled data in supervised learning, Mixup Contrastive Supervised Learning proves to be more robust than both Supervised Contrastive Learning (-2.9%) and Cross Entropy (-4.2%). (3) Our K-NN-based confident samples identification method outperforms the random method by 6.5% and the NLI method by 3.3%. This indicates that our K-NN method can effectively select confident samples.

### 4.6 Analysis on KNN

We conducted an evaluation of the performance of weighted k-nearest neighbors (kNN) in terms of its ability to select confident samples. To elaborate, we intentionally corrupted the labels of instances in the nyt10m test set with a random probability of 20%, 40%, and 60%. Our objective was to assess whether our weighted kNN method could effectively identify the uncorrupted (confident) instances. We trained our model on the corrupted nyt10m test set for 10 epochs, considering its relatively smaller size compared to the training set, which required more epochs to converge. In order to evaluate the ability of the weighted kNN in identifying confident samples, we reported the recall and precision metrics. The results are shown as 4:

It is worth noting that precision is the more important metric because our goal is to make the la-

| | Pre. | Rec. |
|---|---|---|
| 20% | 0.9851 | 0.8970 |
| 30% | 0.9732 | 0.8758 |
| 40% | 0.9531 | 0.8723 |
| 50% | 0.9254 | 0.8655 |
| 60% | 0.8735 | 0.8172 |

Table 4: The effect of KNN.

beled data as clean as possible after selecting confident samples. Even with a noisy label rate of 60%, our weighted kNN method still achieves a precision of 87.35% for the identified confident samples. This indicates that only 12.65% of the labeled data is noisy, which is significantly smaller than the 60% noise rate. Additionally, our weighted kNN method demonstrates a recall of over 80% for confident samples. Although a few clean samples may not be selected, they can still be utilized through pseudo-labeling techniques.

### 4.7 Analysis on Dropout rate

Figure 3 shows the performance of SSLRE under different dropout rate for strong augmentation. we can observe that: (1) Increasing the dropout rate appropriately improves the model's performance, with SSLRE achieving the best result (63.8 on F1) when the strong augmentation dropout rate is set to 0.4. (2) Augmentation using a very high dropout rate harms the performance, as it results in a loss of a significant amount of information. However, this does not significantly degrade the performance since we only use strong augmentation to predict pseudo-labels, which only affects $\mathcal{L}_u$.

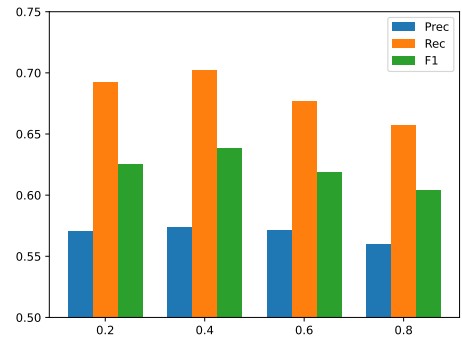

Figure 3: Strong augmentation with different dropout rate

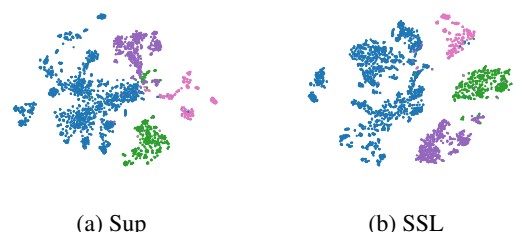

|         |         |
|:-------:|:-------:|
| (a) Sup | (b) SSL |

Figure 4: t-SNE visualization of representations with Pseudo labeling(SSL) and without(Sup). SSL achieves a better cluster results comparing with Sup.

## 4.8   t-SNE analysis

To demonstrate that preserving unlabeled samples can facilitate the learning of a superior representation compared to discarding them, we utilized sentence representations obtained from $theta$ as the input to conduct dimension reduction via t-SNE and acquire two-dimensional representations. We focused on four primary categories of relation classes, which are "/location/location/contains", "/business/person/company", "/location/administrative_division/country", and "/people/person/nationality". As depicted in Figure 4, leveraging unlabeled samples via Pseudo labeling enhances the clustering of identical relation data points and effectively separates distinct classes from one another. Appendix F shows the t-SNE results of all classes.

## 5   Conclusion

In this paper, we propose SSLRE, a novel sentence-level framework that is grounded in Semi-Supervised Learning for the DSRE task. Our SSLRE framework employs mixup supervised contrastive learning to tackle the noise present in selected samples, and it leverages unlabeled samples through Pseudo Labeling, which effectively utilize the information contained within noisy samples. Experimental results demonstrate that SSLRE outperforms strong baseline methods in both sentence-level and bag-level relation extraction.

## Limitations

In order to augment textual instances, we leverage dropout during forward propagation. This necessitates propagating each instance twice to generate the augmented sentence embeddings. However, the demand for GPU resources is higher compared to previous methods. Furthermore, we adjust the dropout rate to regulate the augmentation intensity

for semi-supervised learning and show its effectiveness through the performance results. Nonetheless, we have not conducted explicit experiments to investigate the interpretability, which needs further investigation.

## Acknowledgment

This work is sponsored by National Natural Science Foundation of China (NSFC) (62206291, 62141608, and 62236010).

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

## A Algorithm

Algorithm 1 provides the pseudo-code of the overall framework.

## B Held-out evaluation

| Models | Prec. | Rec. | F1 |
|---|---|---|---|
| SENT | 45.3 | 50.1 | 47.5 |
| HiCLRE | 44.3 | 50.6 | 47.2 |
| SSLRE(Ours) | **49.2** | **54.3** | **51.6** |

Table 5: Held-out evaluation on NYT10

Table 5 shows the held-out evaluation results on NYT10 dataset.

## C Evaluation Settings

**Sentence-level evaluation**: Different from bag-level evaluation used by MIL-based model, a sentence-level(or instance-level) evaluation accesses model performance directly on all of the individual instances in the dataset, which require the model to accurately predict the relation for each sentence. Following (Jia et al., 2019; Ma et al., 2021; Liu et al., 2022), we report micro-Precision($\mu$Prec.), micro-Recall($\mu$Rec.) and micro-F1($\mu$F1) for sentence-level evaluation.

## D Parameter Settings

**Bag-level evaluation**: Bag evaluation accesses the performance of bag relation label extraction. Since manually annotated data are at the sentence-level, following (Gao et al., 2021a), we construct bag-level annotations in the following way: For each bag, if one sentence in the bag has a human-labeled relation, this bag is labeled with this relation; if no sentence in the bag is annotated with any relation,

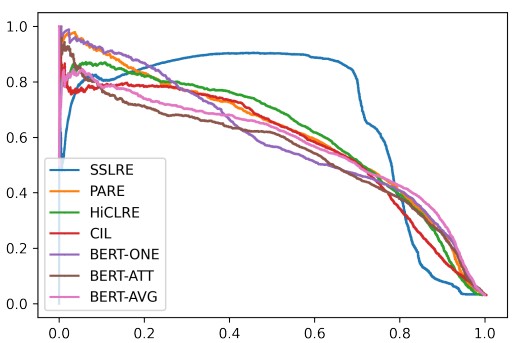

Figure 5: PR-curve

this bag is labeled as N/A. We report AUC, Micro-F1 and Macro-F1 for bag level evaluation.

The underlying encoder for sentence are implemented by BERT-base (Devlin et al., 2019), which generates 768 hidden units for each token's context-aware representation. During the training stage, we set the learning rate of the model to $2 \times 10^{-5}$ and the batch size to 64, which was determined through a grid search over batch sizes in $\{16, 32, 64\}$ and learning rates in $\{1e-5, 2e-5, 5e-5\}$. We train the model for 5 epochs and use Adam (Kingma and Ba, 2014) as the optimizer. The Mixup parameter $\alpha_m$ is set to 1, the classifier loss weight $\lambda_c$ is set to 0.2, the fractile $alpha$ is set to 0.8, the unsupervised loss weight $\lambda_u$ is set to 1, and the CPL threshold $\tau$ is set to 0.95. We set the dropout rate for weak augmentation to 0.2 and the dropout rate for strong augmentation to 0.4. Further analysis on the strong augmentation dropout rate is presented in Section 4.7.

## E PR-curve

We report the P-R curve on NYT10m dataset as Figure 5:

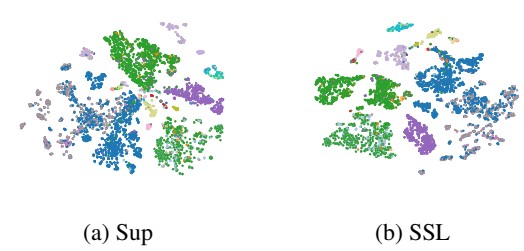

(a) Sup          (b) SSL

Figure 6: t-SNE visualization of representations with Pseudo labeling(SSL) and without(Sup). SSL achieves a better cluster results comparing with Sup, especially on color green and light purple.

## F  Additional t-SNE analysis

Figure 6 shows the t-SNE results on all classes of the sentence representation.

---

**Algorithm 1:** SSLRE Algorithm

---

    **input**   :Noisy Dataset $\tilde{\mathcal{D}}$
    **output :**

1  Warm up $\theta$ and $\phi$ for one epoch using supervised contrastive learning and get $\theta'$.
2  Gain features: $z_i = \theta'(x_i)$
3  Build weighted K-NN graph using Eq. (1)
4  Gain $\mathcal{X}$ and $\mathcal{U}$ using Eq. (2-4)
5  Reinitialize $\phi$ and $\theta$
6  **while** *not reach the maximum iteration* **do**
7     **for** $c \leftarrow 1$ **to** $C$ **do**
8         Calculate $\mathcal{T}(c)$ using Eq. (7) {Determine the flexible threshold for class $c$.}
9     **end**
10     **for** $t \leftarrow 1$ **to** *num_iters* **do**
11         From $\mathcal{X}$, draw a mini-batch $X^t = \{(x_b, y_b); b \in (1, \ldots, B)\}$
12         From $\mathcal{U}$, draw a mini-batch $U^t = \{u_b; b \in (1, \ldots, B)\}$

            `/* Learning from labeled dataset X */`
13         $Z^t = \theta_w(X^t) \cup \theta_w(X^t)$ `// Get two augmented embedding views by dropout inference twice.`
14         Calculate $L^{CLS}$ using Eq. (13).
15         Mixup embedding using Eq. (10), and gain $Z^t_{mix}$.
16         Calculate $L_s$ using Eq. (11)

            `/* Learning from unlabeled dataset U */`
17         Genarate pseudo labels using Eq. (5).
18         Calculate unsupervised loss using Eq. (6) with $\mathcal{T}(c)$.

19         Calculate the overall loss $L$ using Eq. (14).
20     **end**
21  **end**

---