# OpenReview forum: "Noise-Robust Semi-Supervised Learning for Distantly Supervised Relation Extraction"
_EMNLP/2023/Conference — EMNLP 2023 Findings_

### Official Review · Reviewer_JnUq · 2023-07-21

**Soundness:** 4

**Excitement:**

3: Ambivalent: It has merits (e.g., it reports state-of-the-art results, the idea is nice), but there are key weaknesses (e.g., it describes incremental work), and it can significantly benefit from another round of revision. However, I won't object to accepting it if my co-reviewers champion it.

**Paper Topic And Main Contributions:**

The paper addresses the task of distantly supervised relation extraction, and more specifically, assigning relation labels at the sentence level instead of the typically used bag-level. The authors present a semi-supervised learning approach that, in contrast to prior work, does not filter out low-confidence / noisy instances during training, but instead treats them as unlabeled instances that get assigned pseudo-labels during SSL. The approach incorporates a range of further modifications, such as data augmentation with droptout, curriculum learning and mixup supervised contrastive learning, to address the noise inherent in distantly supervised data. Experimental results on the NYT10m and wiki20m indicate that the proposed approach outperforms strong baseline models. The main contributions of this paper is hence an NLP engineering experiment.

**Questions For The Authors:**

- Question A: In L257, you state that you build a “weighted” K-NN graph, but the weights/distance do not seem to be used in equation (2). Is that correct, or did I misunderstand this? Would it make a difference to compute q_c (s_i) as weighted by the cosine similarity (e.g. reduce the influence of outliers)?
- Question B: For Table 1 - how did you arrive at sentence-level results for bag-level methods like RESIDE? In addition, these are explicitly not designed to perform well at sentence-level evaluation since they are trained on bag-level labels only, so I think the comparison is not entirely fair. (I.e. in Table 1 I would only compare to RL-DSRE, ARNOR and SENT). Also - in line 474 you claim that your approach ‘significantly’ outperforms all baselines - if so, please support this with stddev/conf interval info
- Question C: In Table 1, micro-F1 scores for the BERT-XXX are around ~54%, but Gao et al. 2021 report around ~64 micro-F1 for sentence-level eval (Table 4 in their paper, except for BERT-ATT). What is the reason for this drop of 10 points?  Similarly for bag-level evaluation, results reported in Gao et al. for micro-F1 are at 60-62
- Question D: In Tables 1-3, do you report results of a single run, or averaged over multiple runs? (I couldn’t find info on this in the paper, but maybe I overread something)
- Question E: (Suggestion) If possible, could you include P/R curves in the appendix, as is common in DSRE?
- Question F: Do you have an explanation why RESIDE performs so poorly on bag-level evaluation for NYT10m (and to a lesser degree, on Wiki20m)?

**Reasons To Accept:**

- The paper is well organized and mostly easy to follow.
- The approach addresses some interesting gaps in distantly supervised relation extraction
- Overall the study is an interesting contribution to the relation extraction task

**Reasons To Reject:**

- Experimental protocol somewhat under-specified, especially if results are obtained across a single or multiple runs (hindering reproducibility & confidence in reported results)
- Experimental results reported for some of the baselines are inconsistent with results report the original papers (e.g. BERT-XXX, Gao et al. 2021)

**Reproducibility:**

4: Could mostly reproduce the results, but there may be some variation because of sample variance or minor variations in their interpretation of the protocol or method.

**Reviewer Confidence:**

3: Pretty sure, but there's a chance I missed something. Although I have a good feel for this area in general, I did not carefully check the paper's details, e.g., the math, experimental design, or novelty.

**Typos Grammar Style And Presentation Improvements:**

- L014 ‘discarding’
- L017 ‘Relation’
- L36 ‘ONE’ is unclear/unexplained. Is it an abbreviation or an emphasis?
- L79 ‘the DSRE dataset’ - which one - there are many datasets for DSRE, e.g. NYT10,  T-Rex, Dis-Rex
- L192 ‘discussed’
- L193 ‘are not limited to’
- L218 ‘weighted’ ?
- L259 ‘we’
- L291 ‘we’
- L425 - no sentence period, i.e. ‘… of evaluation, we …’ ?
- L431 strategy
- Table 2 - DISTRE seems to be missing - is there a reason for that?

---

> ### Author Rebuttal · Authors · 2023-08-27
>
> We deeply grateful for you detailed suggestions and review. We address your concerns here:
> ##  Q.A: weighted K-nn
>
> Sorry for the equation mistake. **The weights are actually the cosin similarity, which means that the more similar(closer) instance in the K-nerghborhood will contribute more to the $q_c(s_i)$**. So the equation (2) should be: $$\hat{q}\_c(s\_i) = \frac{1}{\sum\_{s\_k \in \mathcal{N}\_i}d(z\_i, z\_k)} \sum\_{ \boldsymbol{s}\_k \in \mathcal{N}\_i}d(z\_i, z\_k) \cdot \mathbb{1}(\tilde{y}\_k = c)$$
>
> The DSRE performance comparison between weighted K-nn and vanilla K-nn on NYT10m is outlined below:
>
> |              | Prec. | Rec. | F1   |
> |:------------:|:-----:|:----:|------|
> | weighted Knn |  **57.4** | **72.0** | **63.8** |
> |  Vanilla Knn |  56.5 | 71.7 | 63.1 |
>
> We also conducted an evaluation of the performance of weighted k-nearest neighbors (kNN) in terms of its ability to select confident samples. To elaborate, we intentionally corrupted the labels of instances in the nyt10m test set with a random probability of 20%, 40%, and 60%. Our objective was to assess whether our weighted kNN method could effectively identify the uncorrupted (confident) instances. We trained our model on the corrupted nyt10m test set for 10 epochs, considering its relatively smaller size compared to the training set, which required more epochs to converge. The other parameter remains consistent with the paper. In order to evaluate the ability of the weighted kNN in identifying confident samples, we report the recall and precision metrics. The results are as follows:
>
> |     |  Prec. |  Rec.  |
> |:---:|:------:|:------:|
> | 20% | 0.9851 | 0.8970 |
> | 30% | 0.9732 | 0.8758|
> | 40% | 0.9531 | 0.8723 |
> | 50%| 0.9254 | 0.8655 |
> | 60% | 0.8735 | 0.8172 |
>
> It is worth noting that **precision is the more important metric because our goal is to make the labeled data as clean as possible after selecting confident samples**. Even with a noisy label rate of 60%, our weighted kNN method still achieves a precision of 87.35% for the identified confident samples. This indicates that only 12.65% of the labeled data is noisy, which is significantly smaller than the 60% noise rate. Additionally, our weighted kNN method demonstrates a recall of over 80% for confident samples. Although a few clean samples may not be selected, they can still be utilized through pseudo-labeling techniques.
>
> ## Q.B
> ### 1)How to arrive at sentence-level results for bag-level methods?
>
> For all multi-instance learning methods(bag-level methods), there always be a sentence encoder to encode each sentence in the bag and a classifier scores $e_i$(or a query based score) representing how well the input sentence matches each relation. Then it aggrates all sentence representation with the guidance of $e$ to get the bag representation. We simply omit the last aggregation step during sentence-level evaluation, just using $e_i$ to get the sentence-level results for inference.
>
> For RESIDE, it also encodes each sentence and aggregates all sentence representation using ATT aggregation method. We omit the ATT aggregation step and use the query score for each sentence to get the sentence-level results.
>
> ### 2) Whether the comparision between bag level methods and sentence level methods using sentence-level evaluation is fair?
> It is essential to emphasize that **the sentence-level labels do not have more information than bag-level labels(that specifies which bag relation label each sentence is corresponding to)**. They are the same actually.
>
> For examples,  <London, United Kingdom> has two relation "/location/location/contains",  "/location/country/capital". Then the bag <London, United Kingdom> is assign with these two label. For sentence level training, each sentence in the bag will be assigned with these two labels(The sentence will occur several times with different label).There is an example from the NYT10m training set as follows. So bag-level labels and sentence-level labels are equivalent, **only the training methods are different**.
> * {"text": "Her husband , Mr. Felgate , 45 , still lives in his native London , where he is a director and spokesman for the Carbon Trust , a publicly funded entity that is working to help the United Kingdom reduce its carbon emissions , as mandated by the Kyoto Protocol .", "relation": **"/location/location/contains"**, "t": {"pos": [59, 65], "id": "m.04jpl", "name": "London"}, "h": {"pos": [181, 195], "id": "m.07ssc", "name": "United Kingdom"}}
> * {"text": "Her husband , Mr. Felgate , 45 , still lives in his native London , where he is a director and spokesman for the Carbon Trust , a publicly funded entity that is working to help the United Kingdom reduce its carbon emissions , as mandated by the Kyoto Protocol .", "relation": **"/location/country/capital"**, "t": {"pos": [59, 65], "id": "m.04jpl", "name": "London"}, "h": {"pos": [181, 195], "id": "m.07ssc", "name": "United Kingdom"}}
>
> Indeed, it is worth mentioning that several other studies, specifically [1] and [2], have also assessed sentence-level results for bag-level methods. Our evaluation methodology aligns with theirs.
> * [1]SENT: Sentence-level Distant Relation Extraction via Negative Training [ACL 2021]
> * [2]Manual Evaluation Matters: Reviewing Test Protocols of Distantly Supervised Relation Extraction[ACL 2021]
>
> ### 3) stddev info
> See Q.D
> ## Q.C inconsistant performance with Gao et al. 2021
> We reproduce Gao et al. 2021's results using their official repository **OpenNRE** with the same hyperparameters as section 5.1 of their paper using seed 42. However, we can not achieve the the sentence-level evaluation results their paper claims. We suspect that there are some  inconsistancy on evaluation code between the scripts they used and the official repository. We used to ask the first author of the Gao et al. 2021 of their original training script, but they lost that. There are also some github issues reporting that they can not reach the  results on the paper. To obtain more rigorous results, we report the average results(seed 1-9) and standard deviation using **OpenNRE**，the results are as follows:
> |                 | NYT10m       |              |              | Wiki20m      |              |              |
> |-----------------|--------------|--------------|--------------|--------------|--------------|--------------|
> |                 | Prec.        | Rec.         | Mircor_F1    | Prec.        | Rec.         | Micro_F1     |
> | Bert-ATT | 49.5 +- 0.46  |58.7 +- 0.55 |53.7 +- 0.52 |71.5 +- 0.61 |78.1 +- 0.45 |74.7 +-  0.51 |
> |Bert-ONE| 52.7 +- 0.95 |62.0 +- 1.12| 57.1  +- 0.96 |73.0 +- 0.49 |73.4 +- 0.28 |73.2 +- 0.33 |
> |Bert-AVG| 52.1 +- 0.31 |55.6 +- 0.32 |53.8 +- 0.31  |80.7 +- 0.28 |76.8 +- 0.25 |78.6 +-  0.27 |
> |                 | AUC          | Micro_F1     | Macro_F1     | AUC          | Micro_F1     | Macro_F1     |
> |Bert-ATT |49.7 +- 0.44 |52.9 +- 0.53 |24.6 +- 0.49| 88.1 +- 0.33 |80.9 +- 0.44|81.0 +- 0.40|
> |Bert-ONE| 57.2 +- 0.53 |54.4 +- 0.82 | 35.8+- 0.71 |89.7 +- 0.36 |81.2 +- 0.21 | 81.9 +- 0.24|
> |Bert-AVG |57.1 +- 0.66 |56.2 +- 0.47 |34.1 +- 0.56 |88.9 +- 0.23 |82.7 +- 0.26|81.2 +- 0.25|
>
> ## Q.D  stddev
> Sorry for not mentioning the relevant experiment protocol. We actually run all experiments of SSLRE using random seed 42. We did not run it 10 times and provide the stddev info because that the results is low-invariance and the seed has a small influence of the results. Due to this, most of the DSRE paper do not provide the seddev info either. However, to be more meticulous，we run SSLRE for ten times with random seed 1-9 and report the stddev info as follows:
>
> |                 | NYT10m       |              |              | Wiki20m      |              |              |
> |-----------------|--------------|--------------|--------------|--------------|--------------|--------------|
> |                 | Prec.        | Rec.         | Mircor_F1    | Prec.        | Rec.         | Micro_F1     |
> | SSLRE_SENT_EVAL | 57.6 +- 0.42 | 71.7 +- 0.51 | 63.8 +- 0.49 | 81.9 +- 0.30 | 81.2 +- 0.24 | 81.6 +- 0.28 |
> |                 | AUC          | Micro_F1     | Macro_F1     | AUC          | Micro_F1     | Macro_F1     |
> | SSLRE_BAG_EVAL  | 66.9 +- 0.43 | 71.6 +- 0.26 | 36.9 +- 0.45 | 91.6 +- 0.23 | 83.5 +- 0.19 | 84.0 +- 0.25 |
>
> As can see, the invariance is small, which means our approach significantly outperforms all baselines is supportable.
>
> ## Q.E P-R curve
> Sure, p/r curve is common in other bag-level DSRE papers. We will report the results on the appendix. But the rebuttal can not provide pictures, we report some important p-r point (the precison when recall are 0.1,0.2...0.9) on nyt10m dataset as follows:
>
> |           |    0.1    |     0.2    |     0.3    |     0.4    |     0.5    |     0.6    |     0.7    |     0.8    |     0.9    |     AUC    |
> |:---------:|:---------:|:----------:|:----------:|:----------:|:----------:|:----------:|:----------:|:----------:|:----------:|:----------:|
> | **SSLRE-ONE** |   0.8091  |   0.8506   | **0.8877** | **0.9028** | **0.9032** | **0.8877** | **0.8008** |   0.3797   |   0.1097   | **66.5** |
> |    PARE   |   0.9133  |   0.8307   |   0.7682   |   0.7273   |   0.6541   |   0.5933   |   0.5056   |    0.408   |   0.2547   |    61.1    |
> |    CLI    |   0.7895  |   0.7831   |   0.7708   |   0.7338   |   0.6563   |    0.584   |   0.5035   |   0.3436   |   0.1599   |    57.4    |
> |   HICLRE  |   0.8647  |   0.8202   |   0.7911   |   0.7632   |   0.7065   |   0.6181   |   0.5157   |   0.3939   |   0.2102   |    61.0    |
> |  RESIDE | 0.7023  |  0.6008 | 0.5560 | 0.5230 | 0.4236 | 0.3120 | 0. 2460 | 0.1350 | 0.0701 | 36.8 |
> |  Bert-ONE | **0.922** | **0.8686** |   0.7697   |    0.665   |   0.5675   |   0.5121   |   0.4659   |    0.408   |   0.2641   |    56.7    |
> |  Bert-ATT |   0.7753  |   0.7136   |   0.6779   |   0.6375   |   0.6171   |   0.5423   |   0.4521   |   0.3787   |   0.2501   |    49.5    |
> |  Bert-AVG |    0.78   |    0.74    |    0.704   |   0.6806   |   0.6375   |   0.5669   |   0.4997   | **0.4258** | **0.2957** |    57.1    |
>
> ## Q.F The poorly performance of RESIDE
> Compaired to other methods, RESIDE performance relatively poorly, we suspect that it is due to the fact that they use entity types as extra information, which leads to overfitting biased heruistics of entities.
>
> ## Other issues:
> * We sincerely apologize for the grammatical errors. We will correct them in the future version.
> * The **ONE** in Line 36 is explained in line 430, which first predicts relation scores for each sentence in the bag and then takes the highest score for each relation.
> * L79, The DSRE dataset are NYT10m and NYT10.
> * DISTRE is missing in the Table2: We apologize for our carelessness, and it should be included in table 2. The results is as follows:
>
> |                 | NYT10m       |              |              | Wiki20m      |              |              |
> |-----------------|--------------|--------------|--------------|--------------|--------------|--------------|
> |                 | AUC          | Micro_F1     | Macro_F1     | AUC          | Micro_F1     | Macro_F1     |
> | DISTRE| 52.1 +- 0.68 |53.1 +- 0.86 |26.6 +- 0.72| 87.1 +- 0.39 |80.9 +- 0.46|81.4 +- 0.44|

---

### Official Review · Reviewer_HoSe · 2023-08-01

**Soundness:** 4

**Excitement:**

4: Strong: This paper deepens the understanding of some phenomenon or lowers the barriers to an existing research direction.

**Paper Topic And Main Contributions:**

This paper focuses on the sentence-level DSRE task and proposes SSLRE framework. SSLRE first selects confident samples with weighted K-NN graph to train the sentence encoder. Then, different from existing methods, SSLRE utilize unselected noisy samples to train the sentence encoder with generated pseudo labels and avoid information loss. Besides, several mechanisms are applied in order to improve performance, such as CPL and mixup.

**Questions For The Authors:**

I’m wondering the performance of the weighted KNN graph to select confident samples. Maybe you can evaluate it with the NYT10m test set, which is manually annotated.

**Reasons To Accept:**

The idea of utilizing noisy samples instead of simply throwing them is novel, and the pseudo labeling method is interesting. Experiments on sentence-level datasets show significant improvements over various existing models.

**Reasons To Reject:**

Several existing mechanisms are applied, such as CPL and Mixup, making the model a little ad hoc and complex.

**Reproducibility:**

4: Could mostly reproduce the results, but there may be some variation because of sample variance or minor variations in their interpretation of the protocol or method.

**Reviewer Confidence:**

3: Pretty sure, but there's a chance I missed something. Although I have a good feel for this area in general, I did not carefully check the paper's details, e.g., the math, experimental design, or novelty.

**Typos Grammar Style And Presentation Improvements:**

There appears several mistake of capital letters, for example Line	14, 47, 222-224, etc.

---

> ### Author Rebuttal · Authors · 2023-08-26
>
> Thank you for providing valuable suggestions and acknowledging our work.  We deeply appreciate your thorough review.
>
> ## 1) The performance of weighted KNN to select confident samples.
>
> We conducted an evaluation of the performance of weighted k-nearest neighbors (kNN) in terms of its ability to select confident samples. To elaborate, we intentionally corrupted the labels of instances in the nyt10m test set with a random probability of 20%, 40%, and 60%. Our objective was to assess whether our weighted kNN method could effectively identify the uncorrupted (confident) instances. We trained our model on the corrupted nyt10m test set for 10 epochs, considering its relatively smaller size compared to the training set, which required more epochs to converge. The other hyper-parameters remains consistent with the paper. In order to evaluate the ability of the weighted kNN in identifying confident samples, we reported the recall and precision metrics. The results are as follows:
>
> |     |  Prec. |  Rec.  |
> |:---:|:------:|:------:|
> | 20% | 0.9851 | 0.8970 |
> | 30% | 0.9732 | 0.8758|
> | 40% | 0.9531 | 0.8723 |
> | 50%| 0.9254 | 0.8655 |
> | 60% | 0.8735 | 0.8172 |
>
> It is worth noting that **precision is the more important metric because our goal is to make the labeled data as clean as possible after selecting confident samples**. Even with a noisy label rate of 60%, our weighted kNN method still achieves a precision of 87.35% for the identified confident samples. This indicates that only 12.65% of the labeled data is noisy, which is significantly smaller than the 60% noise rate. Additionally, our weighted kNN method demonstrates a recall of over 80% for confident samples. Although a few clean samples may not be selected, they can still be utilized through pseudo-labeling techniques.
>
> ## 2）The novelty of our paper.
> It's true that some components are already be proposed by previous papers. However, we also propose other original components for DSRE: such as Pseudo Classifier Head, dropout based pseudo labeling(adjust agumentation strength by adjusting the dropout rate), MixupContrastive Learning.  In addition to these components, our framework also has its own novelty. **We are the first paper utilize semi-supervised learning for sentence-level training**. Besides, previous semi-supervised learning frameworks ignore the remaining noise of the labeled dataset and we are the first paper to make the supervised part noise-robust and **our work achieves a significant enhancements on both sentence-level and bag-level evaluation**.
>
> ## 3) Other issues
> We sincerely apologize for the typos and grammatical errors. We will correct them in the future version.

---

### Official Review · Reviewer_MQ5c · 2023-08-05

**Soundness:** 3

**Excitement:**

3: Ambivalent: It has merits (e.g., it reports state-of-the-art results, the idea is nice), but there are key weaknesses (e.g., it describes incremental work), and it can significantly benefit from another round of revision. However, I won't object to accepting it if my co-reviewers champion it.

**Paper Topic And Main Contributions:**

This paper proposes a new method, SSLRE, for distantly supervised relation extraction to solve the problem of mislabeled instances. The method uses a few learning objectives including mix-up supervised contrastive learning and curriculum pseudo labeling. The results of three widely used datasets appear to be good.

**Reasons To Accept:**

1. The idea is straightforward and the motivation is well explained.
2. The ablation study is quite good, especially for the virtualization of the representations in Fig. 4.

**Reasons To Reject:**

1. The idea of sufficiently using mislabeled sentences as un-labeled data is not new and a similar idea has been explored two years ago in [1].
2. The techniques proposed in this paper are all existing techniques. Therefore, it appears to be a technical report or a workshop paper rather than a research paper in the main conference.  I do not see a lot of technical novelties and insights in this paper.



[1] Zhang, X., Liu, T., Li, P., Jia, W., & Zhao, H. (2020). Robust neural relation extraction via multi-granularity noises reduction. IEEE Transactions on Knowledge and Data Engineering, 33(9), 3297-3310.

**Reproducibility:**

4: Could mostly reproduce the results, but there may be some variation because of sample variance or minor variations in their interpretation of the protocol or method.

**Reviewer Confidence:**

4: Quite sure. I tried to check the important points carefully. It's unlikely, though conceivable, that I missed something that should affect my ratings.

---

> ### Author Rebuttal · Authors · 2023-08-28
>
> We thank you for your comments and feedback. We address your concerns here:
> ## 1) The comparision with the paper "Robust neural relation extraction via multi-granularity noises reduction", zhang et al. 2021.
> We have not read the Zhang's paper before and we apologize for that. Zhang et al. 2021 indeed utilizes the unlabeled samples, and we will cite it and discuss it in the future version of our paper. However, our approach is completely different from zhang et al. 2021 except for the idea using unlabeled samples.
>
> The biggest difference is that our paper is a **sentence-level training framework** aiming at solving the problem that MIL-based techniques are not proficient in precisely mapping each sentence in a bag with explicit sentence labels, while **Zhang et al. 2021 is a MIL-based method**.  Zhang et al. 2021 identifies the confident samples within each bag(by selecting samples closest to the maximum value instance in the bag), which is actually a variant of **ONE** aggregation strategy, the difference is that **ONE** selects the samples with the maximum value but their work selects more within each bag. Therefore, their approach will still be subject to the limitations of the MIL framework. Our method gets rid of the MIL techniques and utilizes weighted K-nn to identify the confident samples, and then we traing our model in a sentence-level way, which is totally different with theirs.
>
> In terms of the semi-supervised learning, SSL framework can be  divided  into two parts, the supervised part and the unsupervised part. Our methods are completely different from their work in this two part:
> * **Supervised part**. Zhang et al. 2021 just calculates the CE loss for the select confident samples in a bag(the normal way of MIL-based techniques). **They ignore that the labeled set still remains noise**. We utilize mixupcontrastive  learning to untilize the labeled samples, which is noise-robust, which can be shown at Table 3.
>
> * **Unsupervised part**.  Zhang et al. 2021 keeps the consistency between the unlabeled instance and the perturbations(add random gaussian noise). We utilize the unlabeled instance by pseudo labeling. Besides, **our methods decouples the generation of utilization of pseudo lables by two parameter-independent head to avoid error amplifying**.  We do not update our classifier $\phi$ with any unlabeled samples to ensure it to generate reliable pseudo labels. Their methods using just one classifier, which may cause error accumulation during the training stage.
>
> Zhang et al. 2021 does not provide code link in their paper and I tried to contact with the first author but it's a non-exsit email address. We will try our best to reproduce their work and compare the results with it in the future version.
>
> ## 2) The novelty of our work.
> We argue that our methods are not all exsiting techniques. We propose
> * **Pseudo Classifier Head and corresponding Gradient Backpropagation Strategy**: The exsiting approaches(fixmatch, flexmatch) generate and utilize pseudo labels with the same head, which may cause training bias and amplifying the model's errors. To mitigate the problem, we propose utilizing a two-classifier model consisting of an encoder $\theta$ with both a classifier head $\phi$ and a pseudo classifier head $\psi$. We optimize the classifier head $\phi$ using only labeled samples and without any unreliable pseudo labels from unlabeled samples. Unlabeled samples are used solely for updating encoder $\theta$ and pseudo classifier head $\psi$. The details can be seen at Figure 2 and Algorithm 1. The ablation study in table 3 illustrates the superiority of our approach.
> * **MixupContrastiveLearning: we combine mixup with contrastive learning to make our supervised learning part noise-robust**. Note that our dataset is textual data, so directly mixup the instance will not applicable. We mixup the embeddings for different instances to get mixed instance. Then we contrast the mixed instance using mixup label(E.q. 11). The ablation study in Table 3 shows the advancement of MixupContrastiveLearning with the normal supervised contrastive learning.
> * **Data augmentation with Dropout and adjust agumentation strength by adjusting the dropout rate**.
> Pseudo Labling usually needs a strong augmentation and a weak augmention in visual field. It is hard for textual data to adjust the agumentation strength, so we propose to adjust dropout rate to generate embeddings of difference agumentation strength. Figure 3 shows that proper-agumentation(0.4 dropout rate) helps the performance of DSRE.
> * **In addition to these components**, our framework also has its own novelty. **We are the first paper utilize semi-supervised learning for sentence-level training**. Besides, previous semi-supervised learning framework ignore the remaining noise of the labeled dataset and **we are the first paper to make the supervised part noise-robust** and our work achieves a significant enhancements on **both sentence-level and bag-level evaluation**.

---

### Meta-Review · Area_Chair_KLhV · 2023-09-24

**Recommendation:** 3

**Metareview:**

The paper presents an approach for making good use of the noisily labeled instances in distant supervision. The results show the effectiveness of the proposed approach. However, one major concern is that the approach appear to be presenting a collection of existing techniques, which makes the current work less attractive as an original research article. It is nevertheless worth sharing with the community.

---

### Decision · Program_Chairs · 2023-10-07

**Decision:**

Accept-Findings

**Comment:**

The paper presents an approach for making good use of the noisily labeled instances in distant supervision. The results show the effectiveness of the proposed approach. However, one major concern is that the approach appear to be presenting a collection of existing techniques, which makes the current work less attractive as an original research article. It is nevertheless worth sharing with the community.